

# Genetic diversity in *Oryza glumaepatula* wild rice populations in Costa Rica and possible gene flow from *O. sativa*

Eric J. Fuchs[1], Allan Meneses Martínez[2], Amanda Calvo[2], Melania Muñoz[2] and Griselda Arrieta-Espinoza[2]

[1] Escuela de Biología, Universidad de Costa Rica, San José, Costa Rica
[2] Centro de Investigaciones en Biología Celular y Molecular, Universidad de Costa Rica, San Pedro San Jose, Costa Rica

## ABSTRACT

Wild crop relatives are an important source of genetic diversity for crop improvement. Diversity estimates are generally lacking for many wild crop relatives. The objective of the present study was to analyze how genetic diversity is distributed within and among populations of the wild rice species *Oryza glumaepatula* in Costa Rica. We also evaluated the likelihood of gene flow between wild and commercial rice species because the latter is commonly sympatric with wild rice populations. Introgression may change wild species by incorporating alleles from domesticated species, increasing the risk of losing original variation. Specimens from all known *O. glumaepatula* populations in Costa Rica were analyzed with 444 AFLP markers to characterize genetic diversity and structure. We also compared genetic diversity estimates between *O. glumaepatula* specimens and *O. sativa* commercial rice. Our results showed that *O. glumaepatula* populations in Costa Rica have moderately high levels of genetic diversity, comparable to those found in South American populations. Despite the restricted distribution of this species in Costa Rica, populations are fairly large, reducing the effects of drift on genetic diversity. We found a dismissible but significant structure ($\theta = 0.02 \pm 0.001$) among populations. A Bayesian structure analysis suggested that some individuals share a significant proportion of their genomes with *O. sativa*. These results suggest that gene flow from cultivated *O. sativa* populations may have occurred in the recent past. These results expose an important biohazard: recurrent hybridization may reduce the genetic diversity of this wild rice species. Introgression may transfer commercial traits into *O. glumaepatula*, which in turn could alter genetic diversity and increase the likelihood of local extinction. These results have important implications for *in situ* conservation strategies of the only wild populations of *O. glumaepatula* in Costa Rica.

# INTRODUCTION

Crop wild or weedy relatives (CWR) are an important source of genetic diversity for modern agriculture (*Hajjar & Hodgkin, 2007*). Genetic variability in these wild populations is important for breeding programs and genetic crop improvement (*Brondani et al., 2005*). The study and conservation of CWR populations should be a priority to secure genetic

Corresponding authors
Eric J. Fuchs, e.j.fuchs@gmail.com
Griselda Arrieta-Espinoza,
griselda.arrieta@ucr.ac.cr

resources for future breeding programs. This is particularly important for critical food staples. Most people on earth use rice (*Oryza sativa*) as their major staple food and it represents 19% of the world's per capita energy consumption (*Maclean, Hardy & Hettel, 2014*). One-fifth of the world's population depends on rice cultivation for their livelihood (The Sustainable Rice Platform, http://www.sustainablerice.org/). Wild rice species are valuable resources for breeding programs. There are 21 identified wild rice species; six of them are diploid and capable of hybridizing with commercial *O. sativa* (*Khush, 1997*; *Vaughan, Morishima & Kadowaki, 2003*). These wild species have been shown to be important sources of novel and commercially important traits, such as tolerance to acid soils and drought and yield improvements (*Brar, 2005*; *Hajjar & Hodgkin, 2007*). *Oryza glumaepatula* is the only diploid ($A^{gp}A^{gp}$) native rice species in America.

Central American populations of *O. glumaepatula* are generally small and fragmented due to specific habitat requirements (e.g., rivers and wetlands). Additionally, land use changes have negatively impacted habitat availability (*Vaughan et al., 2005*), reducing population sizes, which in turn may reduce genetic diversity within populations and increase genetic structure among them. Small, isolated populations are more likely to exhibit lower levels of intrapopulation genetic diversity and higher structure among them, attributes that increase the risk of extinction due to genetic or ecological factors (*Newman & Pilson, 1997*). Mesoamerican populations have not yet been analyzed; genetic diversity studies have primarily been conducted in Brazil (*Karasawa et al., 2007b*). These studies have consistently found a larger degree of genetic diversity structured among populations in different rivers, likely due to limited gene flow (*Akimoto, Shimamoto & Morishima, 1998*; *Buso, Rangel & Ferreira, 1998*; *Karasawa et al., 2007b*; *Veasey et al., 2008*; *Veasey et al., 2011*). Populations within a river system are less likely to show differences in allele frequencies. Seed dispersal through waterways or river systems in hydrochorous species is likely to homogenize genetic diversity among populations. However, the location of populations along the river may influence genetic diversity estimates. Upriver populations are more likely to contribute seeds and propagules to downriver populations, increasing genetic diversity. Upriver populations receive less gene flow and their genetic diversity should decrease because of genetic drift.

Gene flow between related interfertile species may result in gene introgression (*Chu & Oka, 1970*). Introgressive hybridization is thought to play an important role in plant evolution (*Grant, 1981*; *Arnold, 1997*) by introducing novel genetic variation which allow species to exploit new adaptive landscapes (*Ellstrand, Prentice & Hancock, 1999*). However, introgression often increases genetic differences between wild and introgressed populations (*Jiang et al., 2012*) and may pose a threat if hybridization reduces local adaptation (outbreeding depression) by culling autochthonous genetic diversity (*Ellstrand, Prentice & Hancock, 1999*). Introgression from cultivated rice into wild or weedy rice species has been documented in controlled and natural settings (*Song et al., 2003*; *Chen et al., 2004*), and although introgression may lead to a short-term increase in genetic diversity within populations, original or autochthonous diversity is often lost in the process (*Lu, 2013*). Introgression in *O. glumaepatula* is highly relevant to biosafety regulations. Gene flow from cultivated species into wild relatives may also pose a biosafety threat if transgenes

from genetically modified (GM) crops are able to move via gene flow into wild populations (*Snow, 2002*; *Ellstrand, 2003*).

With this study we aim to quantify the distribution of genetic diversity within and among Costa Rican populations of the wild rice species *Oryza glumaepatula*. We also intend to determine whether populations within a single river are genetically structured and whether river direction influences the magnitude of genetic diversity. Finally, we want to determine if there is indirect evidence of gene flow between wild rice and cultivated varieties of *O. sativa*, because commercial rice species are planted in the vicinity of wild rice populations. These results have important implications for *in situ* conservation strategies of the only known populations of *O. glumaepatula* in Costa Rica.

## METHODS

### Study species

*Oryza glumaepatula* L. is a wild rice species distributed throughout Central and South America and the Caribbean (*Vaughan, Morishima & Kadowaki, 2003*). Detailed morphological descriptions of *Oryza* wild species in Costa Rica can be found in *Zamora et al., (2003)*. *O. glumaepatula* grows in flooded areas, marshes, rivers and wetlands in clay or loam soils. It is a perennial, tufted and scrambling grass with a brittle culm near the base of plants. Culms may detach and float creating new populations. Flowering occurs between October and November and is immediately followed by a brief two to three week fruiting episode. This species is likely wind pollinated (anemophily) however, previous reports (Ge et al., 1999 reviewed in *Karasawa et al., 2007a*) suggest autogamous pollination. The amount of vegetative reproduction has not yet been quantified.

### Study sites

We analyzed all known populations of *O. glumaepatula* in Costa Rica. These populations are located in two distinct geographic regions, the first in northwestern Costa Rica (10°57′03.3″N/85°36′59.2″W) in a seasonal wetland in the Guanacaste Province (GU, Fig. 1). During the rainy season the area floods, allowing *O. glumaepatula* individuals to grow and form a small population ($n < 100$) restricted to a small area of less than 1 Ha. The second site is located in the Medio Queso wetland (MQ) in northeastern Costa Rica (11°01′34.1″N/84°40′42.8″W, Fig. 1). This palustrine wetland of approximately 5,000 ha is irrigated by the Medio Queso River (*Jimenez, 2004*) and extends for about 10–20 km in length. Gramineous vegetation is the dominant flora with a few tree species that are adapted to a flooded environment (e.g., *Pachira aquatic*). Along riverbanks, farms with livestock and maize, bean and rice plantations are common. *O. glumaepatula* is commonly found in MQ, with large patches of individuals on both banks of the river and in multiple sites along the MQ wetland. Along the river unconnected patches of >100 specimens are commonly found. Separation between populations may increase during the rainy season when river volume increases (*Jimenez, 2004*).

### Sampling

In GU, we found less than a 100 *O. glumaepatula* ramets in a few separated patches along a marsh (∼2 m deep). Therefore, we were only able to collect 15 specimens (Table 1). In MQ,
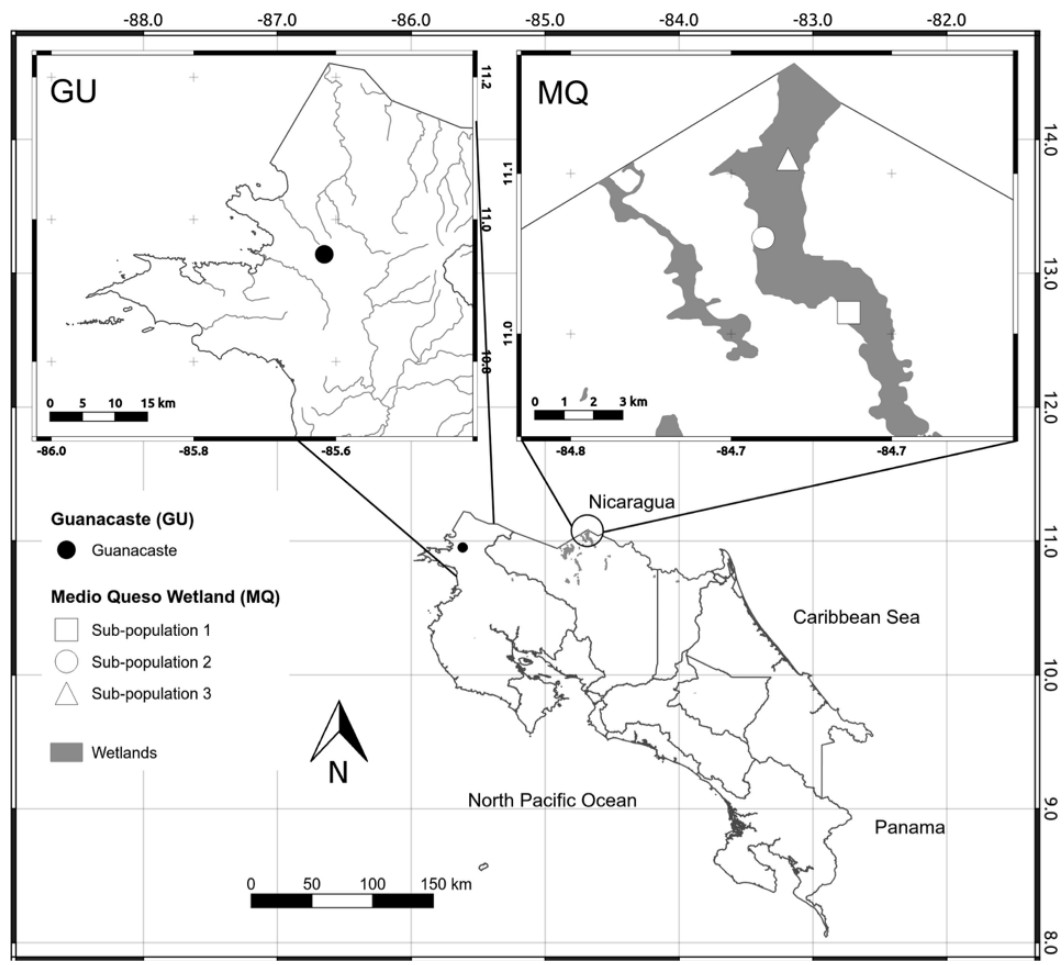

**Figure 1   Map of sampling sites.** Map of Costa Rica depicting the location of the only two known populations of *Oryza glumaepatula*. The insets show the location of the three sub-populations along the Medio Queso (MQ) river and Guanacaste (GU) in Costa Rica.

**Table 1   Genetic diversity estimates for *Oryza glumaepatula* samples from three sub-populations (S1, S2, S3) along the Medio Queso (MQ) river and Guanacaste (GU) in northern Costa Rica.**

| Population | N | I | $H_j$ | P% |
|---|---|---|---|---|
| MQ | 68 | 0.178 | 0.1353 | 81.82 |
| S1 | 17 | 0.160 | 0.1280 | 50.3 |
| S2 | 30 | 0.169 | 0.1326 | 64.04 |
| S3 | 21 | 0.175 | 0.1386 | 62.02 |
| GU | 15 | 0.148 | 0.1196 | 47.27 |

$N$: number of individuals; I, Shannon's index estimated with GenAlEx; $H_j$, Bayesian estimate of expected heterozygosity; %$P$, polymorphic loci $p > 0.05$ estimated with AFLPsurv.

we sampled three sub-populations along the Medio Queso River, starting at the uppermost navigable section of the river with subsequent sub-populations located downriver. Sub-populations were semi-continuous meadows with a few hundred individuals, clearly separated from other sub-populations by approximately one to two kilometers. Within populations, we sampled between 17 and 30 individuals separated by about 10–20 m from each other, in order to avoid collecting genets (Table 1). A boat was used to collect individuals and care was taken to collect the entire individual (including the rhizome). Plants were transplanted to a greenhouse at the Universidad de Costa Rica and kept for further analyses. Additionally, 19 *O. sativa* individuals were analyzed. Seeds from cultivated rice were donated by Centro para Investigaciones en Granos y Semillas (CIGRAS) from Universidad de Costa Rica and represent two varieties "CR1821" and "CR5272," which were released by the Ministry of Agriculture in the early 1970s and have been used for more than 30 years. The Comisión Institucional de Biodiversidad issued permit VI-3150-2010 to Griselda Arrieta-Espinoza authorizing field collections.

## DNA extraction and AFLP genotyping

Total DNA was extracted from 50 to 150 mg of dried leaf material or from 100 to 250 mg of fresh tissue. Flag leaves were used for DNA extraction as it is customary for mature plants since the flag leaf is the last to senesce (*Chen & Ronald, 1999*) The flag leaf of each specimen was collected, dried and grinded using the FastPrep tissue grinder following the manufacturer instructions. Genomic DNA extraction was performed using the FastDNA® (MP Biomedicals, Santa Ana, CA, USA) kit protocol. DNA quantity and quality were determined by gel electrophoresis and via Nanodrop quantification.

Individual genotypes were determined using AFLP following (*Vos et al., 1995*) with modifications indicated by the AFLP kit provided by Applied Biosystems. Based on the AFLP Plant Mapping Kit for small genomes (Applied Biosystems, Inc., Foster City, CA, USA) and the AFLP$^{TM}$ Plant Mapping Protocol 50 ng of genomic DNA were digested with EcoRI and MseI (New England Biolabs, Inc., Ipswich, MA, USA) at 37 °C overnight. Then double-stranded adaptors were ligated to the ends of the DNA fragments, generating template DNA for subsequent PCR amplifications. After incubation, each sample was diluted 20-fold with TE buffer. The ligated adaptors served as primer binding sites for low-level selection in pre-selective amplification of the restriction fragments. The pre-selective amplification mixture was prepared by adding 4 μl of 20-fold diluted DNA from the restriction-ligation reaction, 1 μl of AFLP pre-selective primers (Applied Biosystems), and 15 μl of AFLP core mix. The pre-amplified DNA was diluted again 20-fold with a low TE buffer and PCR for selective amplification were carried out in a 10 μl volume using different fluorescence-labeled EcoRI-primers and MseI primer combinations.

Selective amplification standardization: 32 of the 64 available combinations were randomly used to determine which primers generated the most polymorphic peaks in samples of *O. glumaepatula* and aDNA control from the Applied B iosystemskit; water was used as negative control. The selective amplification of the samples was carried out using these primer combinations twice in independent experiments and in different thermal cyclers (Hybaid and GeneAmp 9700) to assure reproducibility. Based on this

data, eight combinations (E-TC/M-CTC, E-TA/M-CAA, E-AT/M-CAC, E-TG/M-CTA, E-AC/M-CTG, E-AG/M-CAT, E-AA/M-CTC and E-TT/CAC) were chosen for this study.

AFLPs band analysis: amplified fragments were sized on an ABI 3100 sequencer and band scoring was conducted using GeneMarker v.9.1 software. Only AFLP peaks between 150 and 500 bp were considered. The same individual was scored at least three times, to eliminate false positives. We only retained bands that had a consistent intensity (intensity higher than 500), were reproducible and, were separated by at least 2 bp from each other. Bands were scored manually as 1 (present) or 0 (absent) for each fragment within individuals. Two different analysts scored the electropherograms and the consensus between both was recorded as the multilocus genotype for each individual. We combined data from all markers (primer and selective nucleotide combination) for subsequent diversity analyses.

## Statistical analyses

To allow comparison with previous studies, band frequencies and the Shannon diversity index were estimated using GenAlEx 6.5 (*Peakall & Smouse, 2006*). Bayesian estimates of intrapopulation heterozygosity were obtained using AFLPSurv (*Vekemans, 2002*) with non-uniform priors and $F_{IS} = 0.7$ as estimated by *Karasawa et al. (2007b)*. A Spearman correlation analysis was used to determine whether genetic diversity was associated with the position of sub-populations along the river.

Genetic structure and inbreeding coefficients were also estimated using Hickory v1.1 (*Holsinger, Lewis & Dey, 2002*). We selected Hickory because it uses a Bayesian approach to estimate parameters without the need to assume HWE within populations (*Holsinger & Wallace, 2004*). In all cases, default priors, burnin values, and the number of MCMC chains were used. Runs with twice as many simulations yielded comparable results. We compared the full model to a model without inbreeding ($f = 0$), a model with inbreeding ($f \neq 0$) but no genetic structure among populations ($\theta^{II} = 0$). A model where Hickory chooses a random $f$-value from the posterior distribution (i.e., the $f$-free model), while estimating other parameters was also tested. This last model ($f$-free model) may circumvent bias introduced in $f$ estimates from small samples and dominant markers (*Holsinger, Lewis & Dey, 2002*; *Holsinger & Lewis, 2003*). The best model was chosen based on the Deviance Information Criterion (DIC). The model with the lowest DIC was chosen as the most appropriate model. We ran Hickory on two different data sets. The first data set included all sub-populations from MQ and GU. A second data set analyzed all *O. glumaepatula* populations and a sample of 19 *O. sativa* individuals, allowing us to determine genetic structure among cultivated and wild species. Nei's genetic distances were calculated among individuals and used to construct a neighbor-joining tree using the *poppr* (*Kamvar, Tabima & Grünwald, 2014*) and *ape* (*Paradis et al., 2006*) packages in the R statistical language (*R Development Core Team, 2012*).

Genetic structure was also estimated using Bayesian clustering algorithms implemented in Structure v.2.3.4. The most likely number of clusters was estimated for all *O. glumaepatula* sub-populations in MQ and GU. A final analysis included 19 *O. sativa* individuals to determine whether AFLP markers were able to distinguish between commercial and wild species. We selected the admixture model with correlated allele

frequencies to estimate the most likely number of clusters. We discarded the first 50,000 iterations while preserving the subsequent 100,000 iterations for cluster estimation. Previously, we had determined that 25,000 iterations (10,000 burnin) were sufficient for stable $Q$ and $\alpha$ estimates. Multiple runs of STRUCTURE were conducted to test for the most likely number of possible clusters ($K$). We changed the number of clusters from $K = 1$ to $K = 6$ and twenty replications were conducted for each $K$ value. The most likely number of clusters was inferred using STRUCTURE HARVESTER (*Evanno, Regnaut & Goudet, 2005*). We also performed a visual inspection of admixture graphs to assess the likelihood of a single cluster. We used CLUMPP v.1.1.2 to determine the most likely admixture configuration (*Jakobsson & Rosenberg, 2007*) for the most likely cluster number. CLUMPP's default settings and 10,000 chains were selected.

## RESULTS

We analyzed a total of 83 *O. glumaepatula* individuals in MQ and GU. We also resolved AFLP bands on 19 cultivated *O. sativa* individuals (CO). A total of 444 AFLP bands were scored in all individuals. Genetic diversity estimates for *O. glumaepatula* are shown in Table 1. Pooled populations had 84.2% polymorphic loci. Heterozygosity estimates were similar across sub-populations and diversity was not correlated with river position ($r < 0.05, p > 0.05$). Populations located on opposite sides of the Medio Queso River had comparable levels of genetic diversity. GU was the smallest and most isolated, but isolation and population size did not significantly affect genetic diversity. Both Shannon Indexes and Bayesian estimates of heterozygosity were comparable across sites (Table 1).

According to Hickory the full model fit to the data better (DIC $= 5248.19$) than the model without inbreeding (DIC $= 5252.19$) or the model without structure (DIC $= 5643.89$), suggesting a significant departure from Hardy–Weinberg equilibrium (HWE). Average inbreeding was $f = 0.9442 \pm 0.058$ (CI [0.78–0.99]) across *O. glumaepatula* populations (including GU), suggesting a significant heterozygote deficit. Hickory estimated a low but significant genetic structure among sites ($\theta = 0.023 \pm 0.003$). If GU is excluded from the model, genetic structure estimates do not differ significantly. When *O. sativa* individuals are included in the analysis, as expected, there is an increase in genetic structure ($\theta = 0.08 \pm 0.005$), suggesting differences in allele frequencies among wild and cultivated species. Similar results were shown by the neighbor-joining tree (Fig. 2). Commercial rice clustered separately from *O. glumaepatula*. MQ and GU individuals were mixed in separate clades; however, a large proportion of specimens from sub-population three (S3) and GU were clustered in an intermediate clade close to *O. sativa* individuals.

Bayesian structuring algorithms found little evidence of structure among *O. glumaepatula* populations in MQ and grouped individuals into a single cluster ($K = 1$, data not shown). When Guanacaste was included in the analysis, STRUCTURE suggested $K = 2$ as the most likely number of clusters (Fig. 3A). Several individuals in sub-population 3 (S3) and GU were grouped into a second cluster. Including commercial *O. sativa* into the analysis resulted in STRUCTURE grouping individuals into $K = 2$ clusters. However in this case, commercial rice individuals

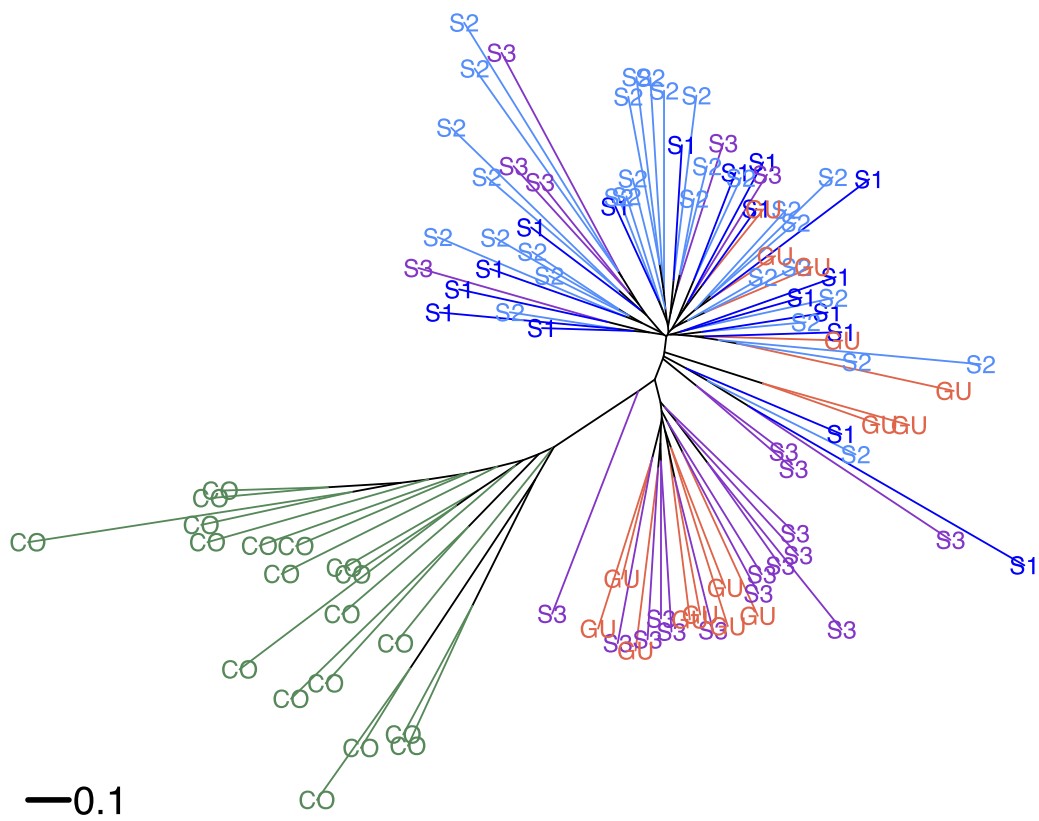

**Figure 2  Neighbour-joining tree.** Neighbor-joining tree showing the relationship between *Oryza glumaepatula* samples from three sub-populations (S1, S2, S3) along the Medio Queso (MQ) river, Guanacaste (GU) and commercial rice (CO) in Costa Rica.

(CO) were assigned to one cluster, whereas *O. glumaepatula* individuals were assigned to a second cluster. STRUCTURE indicates that wild rice individuals at S3 and GU appear to share a significant proportion of their genomes with *O. sativa* (Fig. 3B).

## DISCUSSION

We present the first genetic diversity estimates in Mesoamerican populations of *O. glumaepatula*. Our results suggest intermediate levels of genetic diversity, significant evidence of inbreeding, as well as a lack of structure among populations. Contrary to expectations, we found small differences among populations separated by more than 100 km. We also found indirect evidence of gene flow from cultivated rice in Costa Rican populations of *O. glumaepatula*. These results are of great importance for this CWR; introgressive hybridization may increase the risk of losing autochthonous genetic diversity.

Previous studies on *O. glumaepatula* have shown great variability in intrapopulation diversity estimates (reviewed in *Karasawa et al., 2007b*), which likely respond to differences in population size, connectivity and the nature of markers used to estimate genetic diversity. Generally, studies using hyper-variable markers such as SSR find higher diversity levels (*Brondani et al., 2005*; *Karasawa et al., 2007b*; *Karasawa et al., 2012*). Genetic diversity estimates from Costa Rican populations are comparable to those from Brazil estimated

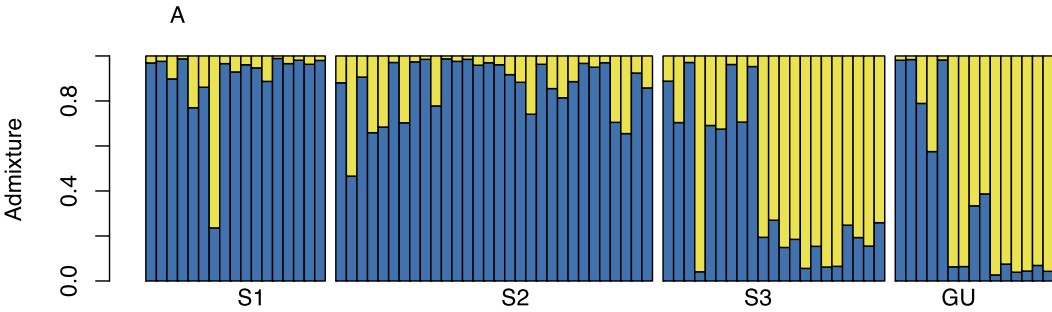

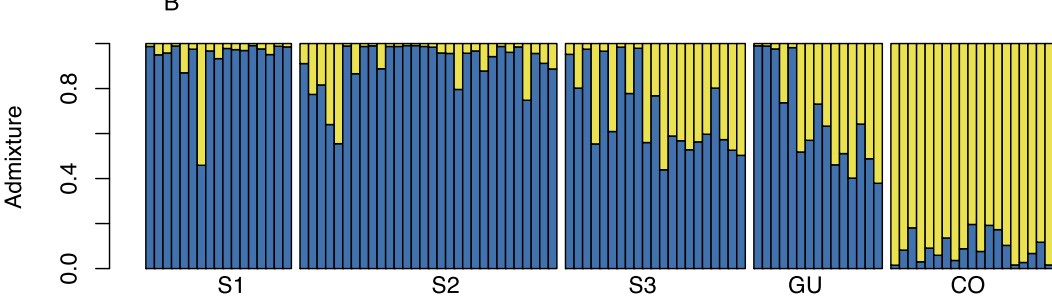

**Figure 3 STRUCTURE admixture graph for MQ and GU.** (A) Admixture of *O. glumaepatula* samples from three sub-populations (S1, S2, S3) along the Medio Queso river (MQ) and Guanacaste (GU) in Costa Rica; assigned to two clusters ($K = 2$) using the Bayesian clustering algorithm in STRUCTURE. (B) admixture of *O. glumaepatula* populations and commercial *Oryza sativa* (CO) individuals.

using co-dominant and dominant markers (*Buso, Rangel & Ferreira, 1998*). This suggests that *O. glumaepatula* populations in Costa Rica have moderate levels of genetic diversity, which are lower than expected for a monocot species with predominant autogamous mating, but similar to those expected for a mixed annual or short-lived species (*Hamrick & Godt, 1996*).

Intrapopulation genetic levels in *O. glumaepatula* are likely influenced by its mating system. As in previous reports we also found a significant deficit of heterozygotes in all populations (*Karasawa et al., 2007b*). As a predominantly selfing species, with a significant proportion of clonal growth, we expect *O. glumaepatula* populations to display high rates of inbreeding. In equilibrium our $F_{IS}$ estimates in MQ and Guanacaste suggest a predominantly selfing mating system ($\hat{t}_m = \frac{1-F_{IS}}{1+F_{IS}} = 0.04$). Progeny arrays and maximum likelihood estimates of *O. glumaepatula*'s mating system obtained comparable results ($tm = 0.01–0.223$. *Karasawa et al., 2007a*). Large interconnected populations are more likely to have higher levels of genetic diversity (*Frankham, 1996*). The *O. glumaepatula* populations in this study are the only populations of this wild rice species found in Costa Rica. These populations are limited to very specific areas (inundated wetlands with a characteristic dry season) and are separated by large distances between each other and from other populations

in Central America or the Caribbean (*Vaughan et al., 2005*). This should increase the effects of drift, reducing intrapopulation genetic diversity. However, MQ is a large metapopulation with more than 10$K$ individuals, all of which are interconnected by the Medio Queso River. This population may be large enough to preserve a significant proportion of the original genetic diversity. Similar results were found for other large and similarly interconnected populations in Brazil (*Buso, Rangel & Ferreira, 1998*; *Vaz et al., 2009*).

We were unable to detect any effect of river direction on genetic diversity estimates. Upstream gene flow is very likely to occur in the MQ river because it does not have a strong current (*Jimenez, 2004*), and during the rainy season floating meadows or seeds may easily move upstream, thus homogenizing genetic diversity estimates across populations regardless of river position. Waterfowl are also likely to disperse seeds among rivers (*Pollux et al., 2007*) and into upstream populations. Populations interconnected via vegetative dispersal, seed-dispersal and pollen flow should have comparable levels of genetic diversity regardless of their position along the river, as observed in our sample.

In a review by *Karasawa et al. (2007b)* they consistently found significant differences in allele frequencies among populations—as expected for an autogamous, short-lived species (*Hamrick & Godt, 1996*). In contrast, we found a significant but dismissible structure ($\theta = 0.03$) among sub-populations separated by less than 15 km within the same river. Migration between sub-populations may occur at the beginning of the rainy season (May), when seeds and culms are likely carried along the river. *Vaz et al. (2009)* reached similar conclusions and did not find evidence of structure among samples collected along the same river separated by ∼10 km. Studies that found structure (*Akimoto, Shimamoto & Morishima, 1998*; *Buso, Rangel & Ferreira, 1998*; *Brondani et al., 2005*; *Karasawa et al., 2007b*) consistently analyzed populations separated by large distances (>100 km), and in most cases in different unconnected river systems. In MQ all sub-populations likely form a single meta-population with ample levels of gene flow among individuals and sites.

An interesting and somewhat contradictory result is that GU, which is separated by more than 100 km from MQ, shows almost no genetic structure with MQ (Fig. 2). Individuals in GU are probably the result of a recent founder effect that most likely originated from MQ. A detailed analysis of the environmental requirements of *O. glumaepatula* in Costa Rica shows that it is only likely to occur in the MQ wetland (*Gastezzi, Martínez & Villareal, 2012*). GU is very small (less than 100 individuals), however, allelic frequencies there do not deviate significantly from those in MQ suggesting that this population may have only been recently founded and genetic drift has not had enough time to reduce its genetic diversity. Guanacaste province has the largest number of rice plantations in Costa Rica and the largest processing plants (*Lomas & Herrera, 1985*). Therefore, cultivated rice from different parts of the country is transported to facilities in Guanacaste. *O. glumaepatula* seeds may have been transported from MQ by animal dispersers or humans in the recent past (*Mack & Lonsdale, 2001*; *Nathan et al., 2008*).

We found indirect evidence of gene flow between *O. glumaepatula* and cultivated rice (*O. sativa*). Our analyses showed that *O. glumaepatula* individuals appear to be admixed with cultivated *O. sativa* (Fig. 3). Introgression between *O. sativa* and *O. glumaepatula* has been previously documented when the species are planted in close proximity (*Brondani et al.,*

*2005*). Cultivated rice and *O. glumaepatula* are sympatric in MQ and in GU. In MQ, farms increasingly invade the pristine wetland and in GU *O. glumaepatula* frequently grows in rice farms. However, we expected limited cross-pollination because the flowering phenology of cultivated rice and *O. glumaepatula* rarely overlap. Hybrids between *O. glumaepatula* and *O. sativa* are male sterile, however stigmas are still receptive and can produce viable seeds (*Yamagata et al., 2010*). Therefore, hybrids are able to backcross to either progenitor, which in turn may result in the introgression of parts of the *O. sativa* genome into *O. glumaepatula*. During our collecting trips we may have inadvertently collected third or forth generation backcrosses as well as wild rice individuals without introgression, all of which are morphologically similar. Currently, genomic analyses are being conducted to confirm whether hybridization is occurring in wild rice populations in Costa Rica.

In Costa Rica, gene flow between cultivated and wild rice species may pose an important biohazard because it allows commercial traits to be transferred into the only populations of *O. glumaepatula* in Costa Rica, with a detrimental impact on autochthonous variation. Our results highlight the importance of interspecific gene flow for both *in situ* and *ex situ* conservation strategies as well as present and future breeding programs. Recently, introgression has become a major focus of interest in regards to biosafety issues, particularly relative to gene flow from genetically modified (GM) organisms into non-GM cultivars and their wild/weedy relatives (*Lu & Snow, 2005*; *Sanchez-Olguin et al., 2009*). Biosafety authorities and regulators may use our results to establish zones of exclusion for the eventual release of GM rice in these areas. The information presented in the present study, should be used to delineate conservation strategies and implement better planting practices to allow for the long-term viability of *O. glumaepatula* populations in Costa Rica. Although wetlands are under international protection by the Ramsar Convention (http://www.ramsar.org/wetland/costa-rica) this site is currently endangered by the construction of a road along the northern border of Costa Rica, which could imperil the largest *O. glumaepatula* population in Costa Rica and a likely stepping-stone between Costa Rican and Nicaraguan wild rice populations.

## ACKNOWLEDGEMENTS

The authors would like to express their gratitude to Dr. Ana M. Espinoza and Dr. Elizabeth Johnson for their valuable contribution to initiate this research; and to Dr. Federico Albertazzi Castro for the general coordination of LAC-Biosafety Project in Costa Rica. In addition, special thanks to Cindy Aguilar & Genuar Muñoz for their technical support and to G Barrantes and J Ross-Ibarra for comments on the manuscript. We would like to thank two anonymous reviewers for improving on previous versions of the manuscript.

### Funding

This work was funded by (1) Latin America: Multi-Country capacity building for compliance with Cartagena Protocol on Biosafety (GEF-UN TP091844) and (2)

Vicerrectoría de Investigación—Universidad de Costa Rica (801-B1-510). The funders had no role in study design, data collection and analysis, decision to publish, or preparation of the manuscript.

**Grant Disclosures**
The following grant information was disclosed by the authors:
Cartagena Protocol on Biosafety: GEF-UN TP091844.
Vicerrectoría de Investigación—Universidad de Costa Rica: 801-B1-510.

**Competing Interests**
The authors declare there are no competing interests.

**Author Contributions**
- Eric J. Fuchs conceived and designed the experiments, analyzed the data, wrote the paper, prepared figures and/or tables, reviewed drafts of the paper.
- Allan Meneses Martínez conceived and designed the experiments, performed the experiments, reviewed drafts of the paper.
- Amanda Calvo and Melania Muñoz reviewed drafts of the paper, scored AFLP's.
- Griselda Arrieta-Espinoza conceived and designed the experiments, contributed reagents/materials/analysis tools, reviewed drafts of the paper.

**Field Study Permissions**
The following information was supplied relating to field study approvals (i.e., approving body and any reference numbers):
Collections were authorized by Comisión Institucional de Biodiversidad permit VI-3150-2010 issued to Griselda Arrieta-Espinoza.

**Data Availability**
Raw data is available at Supplemental Information.

**Supplemental Information**
Supplemental information for this article can be found online at http://dx.doi.org/10.7717/peerj.1875#supplemental-information.

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
