# Peer review of "Genetic diversity in Oryza glumaepatula wild rice populations in Costa Rica and possible gene flow from O. sativa"

_PeerJ, doi:10.7717/peerj.1875_

## Round 0.1 · original submission · Major Revisions

· Academic Editor

Major Revisions

Both reviewers think that the paper may be publishable if major revisions are conducted. Please, follow the suggestions of the reviewers.

Reviewer 2 additionally suggests the incorporation of additional data. Please, address the comments regarding experimental design.

Please, try to improve the English language, perhaps with the help of a native speaker or a commercial language improving service.

Reviewer 1 ·

Basic reporting

Please see General Comments for the Author.

Experimental design

General Comments for the Author.

Validity of the findings

General Comments for the Author.

Comments for the author

This manuscript reports the study of genetic diversity and genetic structure of wild rice (Oryza glumaepatula) populations in Costa Rica. The authors applied AFLP molecular markers to study the wild rice populations. The objectives of this study are to determine whether the wild rice populations along a river are genetically structured and whether river direction plays a role in the magnitude of genetic diversity; and to detect introgression between the wild rice and cultivated rice (O. sativa). Through analyses of the AFLP fingerprints based on 79 O. glumaepatula individuals, the authors reached the following conclusion: 1) Oryza glumaepatula populations in Costa Rica showed a moderate level of genetic diversity and are lack of structure among populations; 2) the position of Oryza glumaepatula populations along the river was not associated with thier genetic diversity; 3) there was evidence of introgression from cultivated rice into its wild rice. This study generated useful information for audience who work in the similar fields although the samples used in this study is small. I recommend publishing this manuscript under the condition that significant improvement should be made for the manuscript. The following points should be considered when the authors revise their manuscript.

1. In general, this manuscript is too long considering the amount of information included. There are a lot of redundant information in the manuscript, which makes it too long and difficult to read. For example, the introduction of Oryza glumaepatula are redundantly presented in both the Introduction and Materials/Methods sections, and the authors included many discussions in the Results section. Therefore, the authors need to substantially reorganize and refine the manuscript, and remove redundancies in the manuscript.
2. The Materials/Methods section is too fragmented. The portion of Study Species is not necessary to be included here because the species has been introduced in the Introduction section. I suggest to remove it. The Study Sites and Sampling portion can be merged into one. The Materials/Methods section needs to be shortened.
3. The samples of individuals from the studied populations are very small (4-22 individuals), compared to the actual population size in nature. As a consequence, the results obtained from the analysis may have considerable errors, meaning the variation pattern may not represent the true variation in the populations. Therefore, when making a conclusion based on the results, the authors need to consider their small samples.
4. The authors included six individuals of cultivated (CV) rice for the STRUCTURE analysis. Can the six CV individuals represent the current and historical variation of CV rice varieties in the vicinity of Oryza glumaepatula populations studied?
5. The authors claimed relatively strongly “introgression from cultivated rice into its wild ancestor…” However, the evidence of introgression from cultivated rice to Oryza glumaepatula is not strong enough from the results of STRUCTURE analyses. Have the authors checked the shared alleles between CV and wild rice? Even the authors find such shared alleles, how could they distinguish the shared alleles is due to their common ancestry, or introgression from cultivated rice into the wild rice? The authors need to explain this. Otherwise, this conclusion is farfetched.
6. The discussion section is too long and unfocused. The authors need to reorganize this section and focus on the results obtained and not to expanded too mach.

Minor points:

1. What does “comparable levels of genetic diversity” (p 10, line 303) mean? Comparable to what?
2. Move all discussions (e.g., p10, lines 309-310, and 319) to the Discussion section.
3. The determination of K value as 2 for STRUCTURE analysis seems odd. The authors may recheck this out.
4. The English language needs substantial improvement.

Reviewer 2 ·

Basic reporting

The article is well written and represents a novel study on the genetic diversity of wild rice populations in Costa Rica. However, the results presented are not sufficient to address the two main objectives stated. The article is publishable if the authors could add more analyses to the data presented.

Experimental design

In general, one of the major problems in the methodology is the sample size differences between populations, which ranged from 4 to 22 individuals after plant mortality in greenhouse. In this matter, more collections in order to reduce sample size differences between populations, if possible, should be performed.

Validity of the findings

Though AFLPs can provide robust data, the two main objectives proposed in the article could be better met if complementary marker data were presented.
To address the first objective regarding population structure, the authors cite published studies were the same question was addressed using microsatellite markers. Given the robustness and quality of data that microsatellite markers provide, and their availability for the species, it would be interesting to complement the AFLP data with microsatellite data to support the results.

Regarding the second objective about introgression from cultivated rice species, chloroplastic DNA sequence analyses in addition to AFLP data should be used in order to provide more robustness to the results.

---

## Round 0.2 · Minor Revisions

· Academic Editor

Minor Revisions

Both reviewers agree that the manuscript has considerably improved. There are still few points of concern from reviewer 1. Once these suggestions have been addressed the manuscript can be considered for publication.

Reviewer 1 ·

Basic reporting

The manuscript entitled “Genetic diversity in Oryza glumaepatula wild rice populations in Costa Rica and possible gene flow from O. sativa” has a considerable improvement after the revision made by the authors. The manuscript now focuses on the level and distribution of genetic diversity within and among populations of Oryza glumaepatula occurring in Costa Rica. Based on the results, the authors reach two main conclusions: (1) Costa Rican Oryza glumaepatula populations have moderate levels of genetic diversity and lack of evident genetic differentiation/structures among populations; (2) pollen/seed-mediated gene flow from cultivated rice (Oryza sativa) to Costa Rican populations of O. glumaepatula is detected. These results can be used as valuable references for conservation of the wild species in the country, as well as for the biosafety assessment of the crop-to-wild gene flow issues.

However, there are still some minor points that affect the quality of the manuscript and should be treated. Therefore, I recommend the publication of this manuscript in PeerJ under the condition that minor revision has been made. The following points should be considered by the authors when modify their manuscript.

1. There are still quite a lot of redundancies in the Discussion section, which makes the manuscript verbiage. The authors should considerably improve this situation.

2. The authors mentioned in their sampling procedures that “flag leaves were used for DNA extraction.” Why it is so important to emphasize the “flag leaves”? Unless there is a special reason to only include flag leaves, it might be sufficient to indicate leaf samples are used for DNA extraction. Otherwise, it may mislead the audience.

3. The authors should provide the full description of the abbreviations (such as HWE in the text and MQ in Table 1) at their first appearance to help the audience to understand the manuscript.

4. The authors should consistently use their abbreviated codes for the study sites (such as GU for Guanacaste) throughout the manuscript. For example, Guanacaste and GU are used inconsistently in the manuscript.

5. The authors found a varied level of crop genetic components in different O. glumaepatula populations. Maybe, it is useful to compare the spatial distances between cultivated rice and their collected O. glumaepatula populations, because a close spatial distance between crop rice and wild rice might result in a greater level of gene flow.

6. Fig. 3 can be deleted (or merged) because Fig. 4 has included nearly all information presented by Fig. 3.

Experimental design

The experimental design is OK.

Validity of the findings

The two conclusions are useful for the readers. (see general comment)

Comments for the author

The manuscript entitled “Genetic diversity in Oryza glumaepatula wild rice populations in Costa Rica and possible gene flow from O. sativa” has a considerable improvement after the revision made by the authors. The manuscript now focuses on the level and distribution of genetic diversity within and among populations of Oryza glumaepatula occurring in Costa Rica. Based on the results, the authors reach two main conclusions: (1) Costa Rican Oryza glumaepatula populations have moderate levels of genetic diversity and lack of evident genetic differentiation/structures among populations; (2) pollen/seed-mediated gene flow from cultivated rice (Oryza sativa) to Costa Rican populations of O. glumaepatula is detected. These results can be used as valuable references for conservation of the wild species in the country, as well as for the biosafety assessment of the crop-to-wild gene flow issues.

However, there are still some minor points that affect the quality of the manuscript and should be treated. Therefore, I recommend the publication of this manuscript in PeerJ under the condition that minor revision has been made. The following points should be considered by the authors when modify their manuscript.

1. There are still quite a lot of redundancies in the Discussion section, which makes the manuscript verbiage. The authors should considerably improve this situation.

2. The authors mentioned in their sampling procedures that “flag leaves were used for DNA extraction.” Why it is so important to emphasize the “flag leaves”? Unless there is a special reason to only include flag leaves, it might be sufficient to indicate leaf samples are used for DNA extraction. Otherwise, it may mislead the audience.

3. The authors should provide the full description of the abbreviations (such as HWE in the text and MQ in Table 1) at their first appearance to help the audience to understand the manuscript.

4. The authors should consistently use their abbreviated codes for the study sites (such as GU for Guanacaste) throughout the manuscript. For example, Guanacaste and GU are used inconsistently in the manuscript.

5. The authors found a varied level of crop genetic components in different O. glumaepatula populations. Maybe, it is useful to compare the spatial distances between cultivated rice and their collected O. glumaepatula populations, because a close spatial distance between crop rice and wild rice might result in a greater level of gene flow.

6. Fig. 3 can be deleted (or merged) because Fig. 4 has included nearly all information presented by Fig. 3.

Reviewer 2 ·

Basic reporting

No Comments

Experimental design

No comments

Validity of the findings

No comments

Comments for the author

The main objectives of this article are to 1- Evaluate the genetic diversity between and within O. glumaepatula populations in Costa Rica 2- evaluate genetic structure within populations distributed along the same river 3- Find indirect evidence about introgression of O. sativa into O. glumaepatula adjacent populations.
The authors used AFLP markers to analyze genetic diversity in all of the known populations of O. glumaepatula in Costa Rica and compare it to the genetic diversity of local grown varieties of O. sativa. The main conclusions presented were: 1- O. glumaepatula populations showed a moderate genetic diversity within populations, and a lack of structure among populations 2- There was an indirect evidence of genetic flow between O. sativa and O. glumaepatula.
My main concerns when I firstly reviewed this paper were the small population size and the marker choice.

Regarding the two first points, the authors responded adequately by increasing the number of individuals for the Guanacaste population and for O.sativa. They also provided suitable reasons for not adding more individuals to the Medio Queso population, but they grouped adjacent subpopulations together, which was more adequate,making the results presented in Table 1 and in the STRUCTURE analyses easier to understand. They also provided acceptable reasons for why they didn’t complement their analyses by adding additional markers, and they softened their statement regarding the introgression from O. sativa, which is more appropriate to the results obtained.
Also, the length of the paper as well as the writing were highly improved.

In general, the results shown in this paper represent a helpful scientific source for the implementation of conservation strategies of O. glumaepatula in Costa Rica.

Taking into account the authors explanations, and the changes made in the paper, I agree to accept the manuscript for publishing.

---

## Author Rebuttal · Round 0.2

November 9, 2015

Dr. Marion Röder
Academic Editor
Dear Dr. Röder,

First of all we would like to apologise for the delay in getting the revised version of the manuscript to you. Reviewers suggested an increase in sample sizes, which required further laboratory work and reanalyses. We hope we have addressed the reviewers comments which have clearly improved the quality of the manuscript. The specific comments raised by the reviewers are address in the following paragraphs.

**The main concern raised by the reviewers focused on sample size per population.**

Natural populations of *O. glumaepatula* in Costa Rica are extremely rare. Populations in Medio Queso River (MQ) are currently under strict protection given that the area is threatened by the construction of a new road. Several environmental agencies are trying to prevent the demise of this population and thus destructive sampling is strictly controlled. Acquiring permits to increase our *O. glumaepatula* sample size at this time would very likely fail or require considerable time. This is the main reason why we used plants growing in the greenhouses at Universidad de Costa Rica for our analyses. Nevertheless, the reviewers main concerns were based on the ability to accurately estimate allele frequencies to determine differences and the likelihood of gene flow among sites and species. Given the lack of genetic differences between closely adjacent populations within MQ river, we grouped populations that were separated by less than 500 meters into three sub-populations within the MQ river. They are almost equidistant from each other along the river, increasing sample size per sub-population to approximately 15-20 individuals, which is comparable to other genetic diversity studies in this species. These sample sizes should provide accurate allele frequency estimates. Given the lack of genetic structure among populations, a Wahlund effect is unlikely to influence our conclusions. The three resulting sub-populations are positioned along the river which still allows us to test the effect of river current on genetic diversity.

We did increase sample sizes for Guanacaste (N=15) and commercial *O. sativa* individuals (N=19). The larger sample sizes did not significantly change our

conclusions, however, estimates are likely to be more accurate.  We redid all of our analysis using these new samples.

**Reviewers were concerned that our conclusions about introgression may have been overstated and that in order to confirm the claim we would require different or additional markers.**

We agree with the reviewers that our results provide only indirect evidence of possible gene flow between *O. sativa* and *O. glumaepatula*, and that further analyses should be conducted to confirm our claims about introgression. However, the use of additional markers is beyond the time and financial scopes of the present project and thus could not be included in this version of the manuscript. Therefore we emphasised genetic diversity analyses, and included additional descriptive analyses such as neighbour-joining trees.  We have also tempered our conclusions about introgression and clarified that our results provide only indirect evidence of gene flow and admixture among both rice species.

**The length and wording of the manuscript needed revision.**

The manuscript has undergone significant rewriting and editing. We have reduced the length of the manuscript considerably. For example, given that Geneland and Structure analyses reached the same conclusions, we eliminated Geneland which contributed only to the length of the manuscript. We checked redundancy throughout the  manuscript and clarified the methods section. The manuscript has been checked by two native speakers and by the Scribendi ([www.scribendi.com](www.scribendi.com)) profesional copyediting service,  to improve its readability.

**Reviewer #1 asked if our sample of commercial rice truly represented current and historical variation of commercial rice varieties in the vicinity of our *Oryza glumaepatula* populations**.

Our commercial rice samples are "CR1821" and "CR5272" which were released by the Ministry of Agriculture in the early 1970s and have been used for more than 30 years. *O. glumaepatula* samples from GU and MQ were collected a few years back before the newly released commercial varieties CFX18 and Palmar18 were widely planted in the country.  We are confident that our *O. sativa* samples represent the historic variation in commercial rice in the vicinity of *O. glumaepatula* and the variability at the time of collection. We have included this reasoning in the manuscript.

We hope we have addressed the major concerns of the reviewers. We uploaded a version with Track-changes, however given that the manuscript underwent significant changes this version is somewhat difficult to read. We also include a "clean" version of the new manuscript.

Please do not hesitate to contact us if you have any further questions.

Sincerely yours,

Eric J. Fuchs on behalf of all authors.

---

## Round 0.3 · accepted · Accept

· Academic Editor

Accept

The current version of the manuscript can be accepted for publication.

Reviewer 1 ·

Basic reporting

I have carefully read the revised manuscript entitled “Genetic diversity in Oryza glumaepatula wild rice populations in Costa Rica and possible gene flow from O. sativa”. The authors have addressed all the questions and comments pointed in the previous review. I satisfied with all the corrections made in the revised manuscript. I therefore recommend the publication of this manuscript in PeerJ.

Experimental design

The experimental design is sound.

Validity of the findings

The findings are significant.

Comments for the author

I have carefully read the revised manuscript entitled “Genetic diversity in Oryza glumaepatula wild rice populations in Costa Rica and possible gene flow from O. sativa”. The authors have addressed all the questions and comments pointed in the previous review. I satisfied with all the corrections made in the revised manuscript. I therefore recommend the publication of this manuscript in PeerJ.